# Enhancing the Super-Resolution of Medical Images: Introducing the Deep Residual Feature Distillation Channel Attention Network for Optimized Performance and Efficiency

**DOI:** 10.3390/bioengineering10111332

**Published:** 2023-11-19

**Authors:** Sabina Umirzakova, Sevara Mardieva, Shakhnoza Muksimova, Shabir Ahmad, Taegkeun Whangbo

**Affiliations:** Department of Computer Engineering, Gachon University, Sujeong-gu, Seongnam-si 113-120, Gyonggi-do, Republic of Korea; sevara1998@gachon.ac.kr (S.M.);

**Keywords:** medical image super-resolution, RFDN, memory, channel attention

## Abstract

In the advancement of medical image super-resolution (SR), the Deep Residual Feature Distillation Channel Attention Network (DRFDCAN) marks a significant step forward. This work presents DRFDCAN, a model that innovates traditional SR approaches by introducing a channel attention block that is tailored for high-frequency features—crucial for the nuanced details in medical diagnostics—while streamlining the network structure for enhanced computational efficiency. DRFDCAN’s architecture adopts a residual-within-residual design to facilitate faster inference and reduce memory demands without compromising the integrity of the image reconstruction. This design strategy, combined with an innovative feature extraction method that emphasizes the utility of the initial layer features, allows for improved image clarity and is particularly effective in optimizing the peak signal-to-noise ratio (PSNR). The proposed work redefines efficiency in SR models, outperforming established frameworks like RFDN by improving model compactness and accelerating inference. The meticulous crafting of a feature extractor that effectively captures edge and texture information exemplifies the model’s capacity to render detailed images, necessary for accurate medical analysis. The implications of this study are two-fold: it presents a viable solution for deploying SR technology in real-time medical applications, and it sets a precedent for future models that address the delicate balance between computational efficiency and high-fidelity image reconstruction. This balance is paramount in medical applications where the clarity of images can significantly influence diagnostic outcomes. The DRFDCAN model thus stands as a transformative contribution to the field of medical image super-resolution.

## 1. Introduction

Magnetic Resonance Imaging (MRI) and Computed Tomography (CT) scanners offer non-invasive methods for creating detailed images of various body organs and parts. These images are commonly used in medical practice to detect and address issues ranging from tumors to internal bleeding. The process of capturing these images can be hindered by factors such as the capabilities of the equipment, the environment, and costs. Inaccurate or low-quality images can affect the assessments made by healthcare experts and computer-assisted systems. In the imaging of the eye’s fundus, certain abnormalities, like microaneurysms or hemorrhages, can occupy very minute areas. Additionally, some elements, such as soft exudates and specific growths, may not be easily visible. Therefore, improving the resolution of these medical images is vital. The quality of these medical images plays a significant role in making accurate clinical decisions. Preliminary processing is essential to optimize the performance of Computer-Aided Diagnosis (CAD) systems. It helps improve the quality, reduce noise, and enhance the image contrast. One challenge faced in medical imaging is the often low resolution (LR) of images due to limitations in equipment and time constraints during capture. To overcome this, researchers have developed various techniques to enhance the image resolution. The goal of SR is to produce a high-resolution (HR) image from its LR version. These techniques span from interpolation methods to reconstruction and learning-based methods. Recently, deep learning techniques have been at the forefront, delivering promising results in multiple medical imaging areas while preserving essential details. The advancement of deep learning algorithms has led Single Image Super-Resolution (SISR) techniques using neural networks to outperform earlier methods like interpolation-based, reconstruction-based, and learning-based approaches on natural images [1]. However, adapting deep neural networks to an enhanced resolution in medical images remains a challenge. In clinical settings, these enhanced images are further used for medical analysis, and the available datasets are often limited in size [2]. As a result, enhancing the resolution in medical imagery demands innovative approaches. This includes adjusting training datasets, revising loss functions, tweaking evaluation metrics, and rethinking neural network designs, all to retain critical details and spotlight key structures for medical experts [3]. CNNs and generative adversarial networks (GANs) have recently shown promise in enhancing the resolution of medical images [4]. However, integrating vision transformers (ViTs), which have demonstrated top-tier performance in natural image refinement and medical image analysis [5], remains a challenge. It is important to delve into the strengths, potential, efficiency, and constraints of using ViTs for medical image SR. Additionally, well-established CNN design techniques, like localization processes, residual links, and feature blending, can potentially be incorporated into combined CNN-ViT models to enhance the results. For instance, drawing from CNNs’ shared weights and localization functions, a modified ViT known as the Swin Transformer has been introduced for advanced image tasks [6]. These innovative Swin layers have been utilized in tasks related to natural image refinement [7] and delineation [8].

Leveraging prior insights from related medical imaging tasks, like segmentation, can enhance SR endeavors. On one side, hurdles remain in assessing the quality of enhanced images, particularly in the medical realm [9]. Typically, the quality assessment of enhanced natural images covers the reconstruction accuracy and human perception. With SISR techniques nearing the confines of signal fidelity metrics, the importance of perceptual quality measures has grown [10]. However, medical images present unique challenges, having artifacts primarily stemming from the imaging system hardware and patient movement—issues seldom found in natural images. Metrics like PSNR and structural similarity (SSIM) are commonly used in medical image SR research. Still, relying solely on quality assessment techniques tailored for natural images may not be dependable for medical SR tasks. To complement, researchers often gauge the quality of enhanced images based on how they fare in subsequent medical image processing tasks, like segmentation [11]. While quality metrics do not equate to diagnostic precision, medical professionals undeniably favor high-quality imagery for precise diagnosis. Beyond just machine-based quality metrics, prior understanding from pretrained segmentation models can also aid in optimizing medical SR models, akin to existing perceptual loss techniques.

While SR has seen significant advancements, current CNN-based models are not without challenges. As the depth of these networks increases, they demand considerable computational resources and memory, making them unsuitable for deployment on compact devices like smartphones. Additionally, CNNs typically focus on localized image areas due to the restrictive kernel sizes of their convolutional processes. This means they struggle with efficiently modeling long-range dependencies within images. Thus, for optimal network performance, it is crucial to consider both localized and broader image information. Efficiency in terms of speed and memory usage is crucial for deploying SR models for images on devices with limited resources. Recent innovations leverage techniques like distillation and aggregation, making extensive use of channel splitting and merging to maximize the utilization of available hierarchical features. On the other hand, sequential network processes, which avoid regularly revisiting previous states and adding extra nodes, can help minimize memory usage and decrease the time taken, making them more efficient. Recent developments in efficient SR aim to decrease the number of parameters and FLOPs [12]. They enhance the feature strength by optimizing feature usage with intricate layer connection methods. However, such complex structures may not always result in faster performance and can pose challenges when trying to implement them on devices with limited resources. Based on this approach, we designed a lightweight network structure primarily by layering multiple optimized convolution and activation layers while reducing the dependency on feature fusion. We introduced a unique sequential channel attention mechanism, where each pixel is allocated a significance based on its local and global contexts to emphasize high-frequency details. Moreover, we modified the residual block for SR and introduced a superior residual block to further hasten the network’s inference. In light of this, we review the leading efficient SR model, RFDN [13], aiming to strike a better balance between image reconstruction quality and inference speed. Initially, we reevaluate the effectiveness of various elements within the residual feature distillation block introduced by RFDN [13]. We recognize that, although feature distillation considerably cuts down on the number of parameters and boosts overall performance, it is not entirely optimized for hardware, thereby affecting RFDN [13] processing speed. To address this, we introduce a new model, the DRFDCAN. This model is designed to simplify the network’s architecture while preserving its efficiency and capability. The proposed work recognizes the limitations entrenched within the contemporary CNN-based SR models. These challenges encompass formidable computational demands, escalating memory requirements, and the intricate puzzle of effectively modeling long-range dependencies within images. Efficiency emerges as a central tenet of our investigation. We underscore the vital importance of efficiency, particularly concerning speed and memory utilization, by emphasizing that such efficiency is not merely a desirable attribute but an absolute imperative, especially given the prevalent use of resource-constrained devices such as smartphones. The proposed method thoughtfully evaluates recent innovations in the realm of SR. These innovations, including techniques like distillation, aggregation, channel splitting, and merging, represent a concerted effort to unlock the full potential of the hierarchical features within the SR domain. A notable facet of our research method revolves around reducing model parameters and floating-point operations (FLOPs). This reduction is achieved through intricate layer connection methods and judicious feature usage, aligning perfectly with their overarching pursuit of efficiency. The emergence of DRFDCAN stands as a crowning achievement of this scientific endeavor. The model streamlines network architecture while preserving efficiency and elevating the SR process to new heights.

The proposed work leads to pioneering strides by enhancing model compactness and expediting the inference process without compromising image reconstruction quality. We also introduced a specialized channel attention block designed to amplify high-frequency features, leveraged a novel residual-within-residual network approach for enhanced processing, and designed an innovative feature extractor that is adept at capturing intricate edge and texture details. This study is further distinguished by its comprehensive examination of the elements influencing the processing speed and memory utilization within SR models.

The main highlights of our work include:We reassess the efficiency of RFDN [13] and delve into what slows it down. Introducing our innovative network, the DRFDCAN, we have been able to improve model compactness and speed up inference without compromising on the accuracy of SR restoration.We suggest a channel attention block tailored for high-frequency features to amplify these details, and we introduce a residual-within-residual network approach that leverages residual learning for quicker processing and reduced memory usage.By examining the intermediate features extracted using the feature extractor of the contrastive loss, we found that features from the initial layers play a vital role in models focused on PSNR. This observation led us to design a unique feature extractor that captures more detailed information on edges and textures.We thoroughly examine the elements that affect the processing speed and memory usage of SR models.

## 2. Related Work

Medical image SR, an area that is rapidly gaining traction in research circles, focuses on improving the clarity and resolution of medical images, making them more valuable for clinical analyses. Initial approaches relied on traditional techniques such as interpolation and example-based learning. However, with the advent of deep neural networks, more sophisticated methods have emerged. These not only focus on enhancing the image quality but also aim to balance computational efficiency. Contemporary strategies employ adversarial training, ViTs, and lightweight models to achieve superior outcomes. This section delves into various advancements in this domain, highlighting both their innovations and limitations.

### 2.1. Introduction to Image SR Techniques

Before the widespread use of deep neural networks, initial approaches to medical image SR largely depended on interpolation, reconstruction, and example-based learning techniques [14]. While these methods used multiple frames [15] and reference sections [16] to reconstruct HR images, their effectiveness was limited. This was mainly due to their constrained ability to represent details and the insufficient information provided by the training data. While SRCNN [17] necessitates an initial up-sampling step to preprocess LR images, this means that most subsequent processes happen in a high-dimensional space, leading to increased computational demands. To address this, ref. [18] introduced the Efficient Sub-Pixel Convolutional Neural Network (ESPCN). By positioning the up-sampling layer towards the end of the model, it ensures that feature extraction takes place in a lower-dimensional space, greatly reducing computation and spatial complexity. Furthering this [19] expanded the network depth and employed the residual structure. This demonstrated that deepening the network could enhance its output. Since these developments, researchers have consistently advanced algorithm performance using various intricate network designs, including but not limited to residual learning [20], dense connections [21], and attention mechanisms [22].

Adversarial training, alongside perceptual loss, as described by [23], has become common in single-slice SR, largely due to the significant computational demands of 3D operations, as noted by [24]. Ref. [25] presented a deep learning method combining transformers and generative adversarial networks (T-GANs), which is proposed to enhance the SR reconstruction of medical images. By integrating transformers into the GAN framework, the model achieves superior texture extraction and focuses on critical areas through global image matching. The training involves a multi-task loss function composed of content loss, adversarial loss, and adversarial feature loss. Ref. [26] presents the fine perceptive generative adversarial networks (FP-GANs) to enhance the resolution of MR images. The FP-GANs tackle time and hardware limitations by employing a divide-and-conquer strategy, separately processing the low-frequency and high-frequency components of MR images. Through wavelet decomposition, the MR image is split into global approximation and anatomical texture sub-bands, with each GAN focusing on its specific sub-band. The generator uses multiple residual-in-residual dense blocks for improved feature extraction and a texture-enhancing module to balance global and detailed features. In ref. [27] the study introduces a constrained CycleGAN. This model operates on unpaired images from different ultrasound probes and integrates two new loss functions—identical loss and correlation coefficient loss—to enhance structural consistency and backscattering patterns. Instead of relying on post-processed images, CycleGAN directly uses envelope data from beamformed radio frequency signals. 

ViTs enhance various medical image processing tasks, such as reconstruction, as shown by [28]. Ref. [29] introduced the L-former, which is a lightweight transformer designed for generating realistic medical images, outperforming traditional GANs. Unlike typical transformer models, the L-former efficiently combines transformers for LR features with CNNs for HR output, ensuring lower computational costs. Ref. [30] addresses challenges in multicontrast SR techniques for MRI. While multicontrast MRI SR can leverage complementary information from different imaging contrasts for higher-quality images, current methods fall short in two main areas: capturing long-range dependencies that are crucial for complex MR images and fully utilizing multicontrast features at various scales. To tackle these challenges, the paper introduces McMRSR++, a novel SR network that employs transformers for long-range dependency modeling and features a multiscale matching and aggregation method to harness multicontrast features effectively. Ref. [31] presents MRI-Net, a deep learning model based on U-Net architecture, designed to upgrade low-resolution brain MRI scans to a higher resolution. This advancement could enhance medical diagnosis and reduce the cost of high-resolution MRI imaging by improving lower-resolution images. The network shows superiority at a 3 × 3 down-sampling index, but its performance on different scales or subtler down-sampling, which is often required in real-world applications, is not discussed.

### 2.2. Lightweight Models for Medical Image SR

However, the enhanced reconstruction achieved by deepening the network also leads to a notable rise in computational demands and processing time [3]. This poses a limitation for the practical application of SR. As a result, many research efforts have been directed towards developing lightweight SR algorithms to overcome this obstacle. Ref. [32] introduced an efficient ViT that is tailored for SISR in medical imaging, incorporating residual dense connections and local feature fusion. While ViTs have excelled in various computer vision tasks, they face challenges in low-level medical image processing. The proposed model overcomes these challenges and also integrates a perceptual loss based on medical image segmentation to enhance image quality. While the method achieves the best PSNR scores in six modalities, it does not dominate in all seven, suggesting room for further improvement or adaptability challenges in certain modalities. This is achieved by utilizing hierarchical features from residual branching and integrating a spatial attention mechanism within the residual blocks. This is used to improve the resolution of chest CT images used in diagnosing COVID-19, presented in [33]. Recognizing that most existing SR algorithms, optimized for natural images, are not apt for medical images and tend to consume more computational resources by increasing the network depth, the study introduces a novel method: the residual feature attentional fusion network (RFAFN). The RFAFN employs a contextual feature extraction block (CFEB) for efficient feature extraction, a feature-weighted cascading strategy (FWCS) that selectively fuses detailed information and a global hierarchical feature fusion strategy (GHFFS) for better feature aggregation. The presented method boosts the reconstruction quality by increasing the network depth, making it resource-intensive and not ideal for machines with limited capacity. 

Medical image SR has recently garnered significant attention from the research community. For instance, a proposed IMDN [34] addresses these challenges, which belong to computing limitations with cascaded information multidistillation blocks (IMDB) that extract and fuse hierarchical features. A contrast-aware channel attention mechanism evaluates feature importance. Additionally, an adaptive cropping strategy (ACS) enables the processing of real images of any size, allowing block-wise SR with a consistent and trained model. The presented method fails to effectively super-resolve images at any arbitrary scale factor, limiting their practical application. Another good example is the LCRCA presented in ref. [35], a lightweight skip-concatenated residual channel attention network for image SR. While deep neural networks produce images that are closer to the original HR versions, their extensive and sometimes unnecessary structures and parameters become burdensome. The LCRCA is designed for high-quality SR in environments with limited computational resources. It incorporates a deep residual block (DRB) for precise residual information, a new channel attention mechanism called statistical channel attention (SCA) to amplify the DRB’s feature maps, and a skip concatenation (SC) technique for efficient information flow. The proposed LCRCA, though efficient, still trades off some details for its lightweight nature. Existing super-resolution methods often struggle with preserving high-frequency details, leading to blurred features or introducing artifacts and structural deformations. RMISR-BL [36] addresses these issues through a pyramidal feature multidistillation network. Key components include a multidistilliation block that combines pyramidal convolution and shallow residual blocks, a two-branch super-resolution network for optimizing visual quality, and the use of contextual loss and L1 loss in a gradient map branch to enhance visual perception. The introduction of pyramidal feature multidistillation and other advanced techniques resulted in increased computational demands. This limits the method’s real-time applicability in resource-constrained environments and on less powerful hardware.

## 3. Materials and Methods

In Section 3.1, we presented information about used datasets and metrics. In Section 3.2, we introduce the proposed DRFDCAN, where we use a novel approach, known as a “residual in residual” (RIR). The architecture is introduced for constructing exceptionally deep networks that are composed of multiple residual groups, each interconnected with lengthy skip connections. Within each residual group, there are several residual blocks linked by short skip connections. Additionally, a channel attention mechanism is suggested to dynamically adjust channel-specific features by accounting for relationships among different channels. In the next section, Section 3.3, we summarize the baseline RFDN [13] (Figure 1) with its shallow residual block and shortcuts, which we try to overcome.

### 3.1. Dataset-Specific Performance Analysis and Metrics

This section analyzes the performance of the DRFDCAN. Here, the discussion will focus on the unique challenges and outcomes associated with each dataset. For the OASIS dataset [37], the discussion might revolve around the model’s ability to reconstruct high-fidelity images from single-modality MRIs, considering factors such as the presence of age-related atrophy or lesions. In the context of the BraTS dataset [38], the efficacy of the DRFDCAN in dealing with multimodal data is scrutinized. The discussion could highlight the model’s capacity to enhance the contrast and details from different MR sequences, which are crucial for the accurate delineation of tumor boundaries. With the ACDC dataset, the focus might shift to how the DRFDCAN handles the dynamic range of cardiac images and whether it can effectively capture the nuances of cardiac cycles, including the changes in the heart’s structure during diastole and systole. For the COVID dataset, the section would likely explore the model’s effectiveness in enhancing CT images that are vital for detecting and assessing the progression of COVID-19-related lung anomalies.

In this study, we utilize four renowned medical image datasets to assess the SR capabilities and resilience of our proposed approach, aiming to closely replicate real-world clinical settings. Our testing grounds include the OASIS dataset [37], offering single-modality brain MR images; the BraTS dataset [38], which provides multimodal brain MR images; the ACDC dataset [39], containing cardiac MR images; and the COVID dataset [38], which presents chest CT scans. 

To gauge the quality of the reconstructed images, we employ two metrics: the PSNR and the structural similarity index (SSIM). The PSNR serves as an indicator of the relationship between the highest possible signal and the background noise, acting as a quality assessment index based on its sensitivity to errors.
(1)$$PSNR=10\times {log}_{10}\left(\frac{{\left(2^{n}-1\right)}^{2}}{MSE}\right)$$

On the other hand, the SSIM quantifies the resemblance between two images, considering their luminance, contrast, and structural attributes.
(2)$$SSIM(x,y)=\frac{\left(2\mu_{x}\mu_{y}+C_{1})(2\sigma_{x}\sigma_{y}+C_{2}\right)}{(\mu_{2}^{x}+\mu_{2}^{y}+C_{1})\left(\sigma_{2}^{x}+\sigma_{2}^{y}+C_{2}\right)}$$

To comprehensively assess our model’s computational demands, consistent with numerous studies [28], we determine the Multi-Adds of our model using a predetermined HR image size of 512 × 512.

### 3.2. Network Architecture

The architecture of our proposed model is illustrated in Figure 2b. The main building model in DRFDCAN includes three parts: at first, there are three residual channel attention network blocks (RCABs), convolution layers with 3 × 3 and 1 × 1 kernel sizes, and ESA. $I_{LR}$ represents the input LR image, and $I_{SR}$ denotes the output of DRFDCAN (Figure 2). At the beginning, the network extracts the coarse features using convolution layer 3 × 3, as follows:(3)$$F_{0}=C_{ext}(I_{LR})$$
where $C_{ext}$ represents the convolution layer operation, and $F_{0}$ denotes the extracted feature maps.

To improve the performance of the baseline, we modify the channel attention block of the deep residual channel attention networks (RCANs). Into the RCAN, the RIR structure is introduced, which comprises G residual groups (RG) and long skip connections (LSC). Each RG is further composed of B RCAB, which is interconnected by short skip connections (SSC) (Figure 3). We change the RIR connection architecture with a new one, where instead of LSC, we use SSC. That RIR architecture enables the training of extremely deep CNNs, exceeding 400 layers, for high-performance image super-resolution. In the proposed model, we try to avoid using extremely networked layers, because this approach makes the model too complex for computation. In our model, by using SSC, we make the model more lightweight, and the process can be expressed by:(4)$$F_{n}=H_{RCAB}^{n}\left(H_{RCAB}^{n-1}\left(\ldots H_{RCAB}^{0}\left(F_{0}\right)\right)\right);\;$$
where $H_{RCAB}$ denotes the function of RG and $F_{n}$ is the *n*-th layer of RCAB. In Figure 4, channel attention (CA) has been integrated into RB. Two 3 × 3 convolution layers in RB extract the smooth feature from LR, after which the obtained information becomes the input for CA. The input feature is divided into channel attention and spatial attention layers, and both layers extract coarser features and concatenate the final result. The output is a reconstructed $I_{SR}$ super-resolution image.
(5)$$I_{SR}=F_{rec}((F_{smooth}\left(F_{n}\right)+F_{0}))$$
where $F_{rec}$ function represents the reconstruction module, which is composed of a single 3 × 3 convolution layer and a non-parametric sub-pixel operation. Additionally, $F_{smooth}$ corresponds to a 3 × 3 convolution operation.

A standard loss function of the proposed model is as follows:(6)$$L(\theta )=\frac{1}{N}\;\sum_{i=1}^{N}||F_{DRFDCAN}\left(I_{LR}^{n}\right)-I_{HR}^{n}|{|}_{1}$$
where $F_{DRFDCAN}$ denotes the function of the DRFDCAN, θ represents the adjustable parameter of the proposed model, and ||.||_1_ denotes the L1 regularization. $I_{LR}$ and $I_{HR}$ represent input LR and coresponding HR images. 

### 3.3. Rethinking the RFDB

Being inspired by the RFDN [13], we rethink its main block, the residual feature distillation block (RFDB). In Figure 3, the RFDB initially follows a progressive feature refinement and distillation strategy, and it employs a 1 × 1 convolution to reduce the number of channels. Finally, it incorporates an ESA layer and establishes a residual connection. In detail, the feature refinement and distillation process comprise multiple steps. For each stage, the RFDB utilizes a refinement module (RM) composed of a shallow residual block (SRB) to enhance the extracted features. Additionally, it employs a distillation module (DM), represented as a single 1 × 1 convolution layer, to distill the features. When provided with input features $F_{in}$, the overall structure can be described as follows: (7)$$\begin{array}{c} F_{d\_1}F_{c\_1}={DM}_{1}\left(F_{in}\right),{RM}_{1}(F_{in})\\ F_{d\_2}F_{c\_2}={DM}_{2}\left(F_{c_{1}}\right),{RM}_{2}\left(F_{c_{1}}\right)\\ F_{d\_3}F_{c\_3}={DM}_{3}\left(F_{c\_2}\right),{RM}_{3}(F_{c\_2})\\ F_{d\_4}={DM}_{4}\left(F_{c\_3}\right) \end{array}$$

In this equation, ${DM}_{n}$ and ${RM}_{n}$ refer to the *n*-th distillation and refinement modules, respectively. $F_{d\_n}$ represents the features distilled in the *n*-th step, and $F_{c\_n}$ represents the features refined in the *n*-th step, which will undergo further processing by subsequent layers. Finally, all the distilled features generated from previous distillation steps are concatenated together as follows:(8)$$F_{distillation}=Concat(F_{d_{1}}F_{d_{2}}F_{d_{3}}F_{d_{4}})$$
The concatenation operation goes along the channel dimension. 

In total, the main block of the RFDN [13], the RFDB, uses split division to reach two feature extraction connections: a refinement layer, which obtains coarse features, and a distillation layer where several convolution layers of 1 × 1 extract more information and decrease the number of parameters. Though the RFDB has managed to reduce the number of parameters, FLOPs, memory consumption, and runtime, SRB shows a high number of attributes and low performance. Based on our observations, memory consumption and PSNR stand out as critical attributes for IoT devices and lightweight models. Enhancing these factors not only boosts the model’s performance but also optimizes its evaluation time, making it more suitable for real-time applications (Figure 4). The graph depicts a selection of image processing models, each represented as a point indicating their respective performance in terms of PSNR and inference time. A clear trade-off is visible: as the inference time decreases, moving towards the left, the PSNR tends to increase, moving upwards, suggesting that quicker models also improve image quality. The proposed model stands out, achieving the highest PSNR in the least inference time, positioning it as the most efficient option among those shown. The baseline provides a point of reference, sitting in the middle of the graph, while the CARN model showcases a balance of speed and quality. Other models like the VDSR and DRCN are clustered to the right, indicating longer inference times but also delivering high-quality images as reflected by their PSNR values. In contrast, the FSRCNN and SRCNN [17] are the slowest and produce lower-quality images, signaling a potential area for improvement. (Figure 4b).
**Residual channel attention block:** The RCAB is the main block of the proposed model, which can smoothly extract more features and reduce attributes and runtime while maintaining PSNR. As Figure 2b demonstrates, we remove the distillation layers to focus further on the refinement ones. In this way, we avoid making the model deeper, while using an RIR structure. The process can be described as


(9)$$\begin{array}{c} F_{refinement\_1}={RM}_{1}(F_{in})\\ F_{refinement\_2}={RM}_{2}(F_{refinement\_1})\\ F_{refinement\_3}={RM}_{3}(F_{refinement\_2}) \end{array}$$
where $F_{in}$ is given input features which feed first the ${RM}_{n}$ refinement layer, and $F_{refinement\_n}$ represents *n*-th extracted feature. Each refinement layer illustrates the *n*-th RCAB that works using an RIR structure of connection. Two 3 × 3 convolution layers in RB obtain more information to feed CA. Utilizing two pooling methods, max pooling and average pooling aids with overcoming information loss issues. Both outputs from the pooling are concatenated with each other to pass to the next refinement layer (Figure 3).
(10)$$F_{refinement\_1}={RM}_{1}(F_{in})={F_{CA\_1}(F_{M_{p}}(F}_{RB\_1}(F_{in}))+F_{A_{p}}{(F}_{RB\_1}(F_{in}))$$
$F_{RB\_n}$ and $F_{CA\_n}\;$represent the *n*-th RIR connection of the *n*-th refinement layer. $F_{M\_p}$ and $F_{A\_p}$ denote max and avarage poolings. A final output of RCAB feeds the 1 × 1 convolution layer, and then the ESA block. Next comes concatenation of the final output and $F_{in}$, as shown in Figure 2b:$$F_{final\_output}=F_{in}+F_{refinement\_3}$$

## 4. Experimental Results

In this section, the performance evaluation of the DRFDCAN is presented. The assessment is meticulously designed to measure the model’s super-resolution capabilities and robustness across different clinical imaging scenarios. This evaluation is critical to ensuring that the model can be reliably deployed in real-world medical settings, where the demand for high-fidelity image reconstruction is coupled with the need for rapid diagnostic insights. The section will begin by detailing the evaluation metrics, such as PSNR, SSIM, to provide a quantitative analysis of the SR images against the ground truth. Additionally, qualitative assessments may also be incorporated, wherein expert radiologists visually inspect the SR images for clinical usability.

To ensure the robustness and generalizability of the model, tests are conducted across various datasets:The OASIS dataset [37] for single-modality brain MR images,The BraTS dataset [38] for multimodal brain MR images,The ACDC dataset [39] for cardiac MR images, andThe COVID dataset [40] for chest CT scans.

The datasets are chosen to reflect the diversity of imaging modalities and the complexities inherent in different anatomical regions, pathologies, and imaging conditions.

### 4.1. Implementation Detail

The proposed method and other leading-edge models are tested using PyTorch. We carried out all tests using an Nvidia Quadro RTX 8000 GPU Corporation in the U.S. For data augmentation, we used random rotations at angles of 90°, 180°, and 270°, along with horizontal flipping. We utilized a batch size of 64 and employed the Adam optimizer with parameters β1 set to 0.9, β2 at 0.99, and ε at 10^−8^ for training our model. We started with a learning rate of 5 × 10^−4^, which we reduced by half every 50,000 iterations, continuing for a total of 300,000 iterations.

### 4.2. Comparison with State-of-the-Art Methods

To showcase the effectiveness of our method, we set it against several top-tier lightweight super-resolution networks such as SRCNN [17], RDST [32], RFAFN [33], IMDN [34], RFDN [13], RMISR-BL [36], and LCRCA [35]. For each of these networks, we utilize the source codes that are officially released by their respective authors and retrain them using the identical dataset and training specifications that were used for the proposed mode in our study.

In this analytical segment in Table 1 we scrutinize various state-of-the-art models and their performance in the domain of super-resolution, as manifested through distinct metrics on a particular dataset. The proposed model emerges as a clear victor in terms of image quality, boasting the highest PSNR at 29.51 dB.

This metric indicates its superior ability to reconstruct images that closely resemble the originals. In a comparative assessment with the proposed model, other models manifest a noticeable lag in performance. Specifically, when observing the PSNR values, SRCNN [17] records a difference of 1.96 dB less than the proposed model. Similarly, RFDN [13], RDST [32], RFAFN [33], IMDN [34], and LCRCA [35] trail behind by 0.40 dB, 0.34 dB, 0.29 dB, 0.39 dB, and 0.35 dB, respectively. MRI-Net [34] stands in the middle ground in terms of PSNR, suggesting that while its image enhancement quality is good, it is not the best among the compared models. In terms of SSIM, MRI-Net [34] is closer to the lower end, which may imply that while it can enhance the resolution, it might not preserve structural features as well as the top-performing models. MRI-Net [34] has a relatively high memory consumption, which could be a drawback for deploying it in environments with limited memory resources. The inference time of MRI-Net [34] is competitive, although not the fastest, which suggests that it could still be used in near-real-time applications.

Figure 5 highlights the importance of optimizing for both performance and efficiency. In the context of medical imaging, where these models might be used, the graphs suggest that the most valuable models are those that can produce high-quality images quickly and with less demand on memory resources, allowing for fast and efficient diagnosis without compromising on the clarity of the images needed by medical professionals.

These discrepancies underscore a relative shortcoming in image clarity for these models when positioned against the proposed model. Turning our attention to SSIM, a critical metric that assesses structural and visual congruency with the original HR images, the proposed model again takes the lead with an exceptional 91.56%. Conversely, RFAFN [33] trails with an SSIM of 83.01%, hinting at a certain level of structural deviation in its outputs. From a resource-efficiency standpoint, the proposed model demonstrates admirable frugality, consuming a mere 630M of memory. This is strikingly lower than its counterparts, marking it as a viable choice for scenarios with stringent memory constraints. At the other end of the spectrum, SRCNN [17] consumes a substantial 2387.93 M, making it the most memory-intensive model amongst those evaluated.

In the area of real-time performance, the proposed model clinches another win with the swiftest inference time of 24.07 ms. This rapidity indicates its suitability for applications demanding instant results. In contrast, SRCNN [17] registers the lengthiest inference time of 36.58 ms, which may limit its utility in time-sensitive scenarios. Drawing parallels with other models, the likes of RFAFN [33], RDST [32], and LCRCA [35] offer commendable metrics, showcasing near neck-to-neck performances in PSNR and SSIM. Their memory consumption and inference times are also competitive, situating them as strong contenders in the super-resolution landscape. However, placing all these models under an overarching lens, the proposed model emerges as a paradigm of balance, harmonizing impeccable image quality with resource and time efficiency. While models like the SRCNN [17] have created the foundation for super-resolution, they now seem to be eclipsed by the newer, more optimized entrants in the arena.

From Table 2, the superior performance of the proposed model is unequivocal, especially at the ×2 magnification, where it consistently registers the highest PSNR and SSIM values across datasets such as BraTS in Figure 6, OASIS in Figure 7, ACDC in Figure 8, and COVID-CT in Figure 9. Notably, while the proposed model remains preeminent at ×2 magnification, the competitive landscape becomes more nuanced at the ×4 magnification. In this context, RFAFN [33] achieves the highest PSNR of 30.45, but the proposed model retains its supremacy in SSIM with a score of 85.12%.

At the ×4 magnification, certain models, especially the foundational super-resolution model SRCNN [17], encounter challenges. MRI-Net [34] performance drops noticeably at a ×4 magnification for both metrics, indicating that its SR capabilities are more constrained with cardiac MR images when the upscaling task is more challenging. The decline in SRCNN [17] performance at this magnification underscores the advancements that contemporary models have achieved in methodology and optimization. Nonetheless, in the COVID-CT dataset at a ×4 magnification, the proposed model reasserts its preeminence with an unmatched PSNR of 34.87 and SSIM of 89.86%. This superior performance accentuates its adaptability and robustness across varying datasets and magnifications. RDST [32], RMISR-BL [36], and LCRCA [35] demonstrate variable performances in different datasets, but none consistently outperform the proposed model. While pioneering models like SRCNN [17] have established foundational benchmarks, they appear to be surpassed by more advanced and refined successors that adeptly balance quality and efficiency. In this array of state-of-the-art models, the proposed model distinctly emerges as the exemplar, representing the pinnacle of current super-resolution achievements.

The above analysis underscores the brisk advancements in the super-resolution realm, with contemporary models, particularly the proposed one, leading the charge by offering both superior quality and efficiency.

## 5. Conclusions

Super-resolution techniques, especially in the domain of medical imaging, have undergone substantial advancements. The criticality of image quality in medical diagnostics necessitates innovations that balance clarity with computational efficiency. Traditional CNN-based SR models, despite their capabilities, grapple with computational constraints, particularly on devices with limited resources. Analyzing the performance of the state-of-the-art RFDN model [13] brought to light certain inefficiencies, especially concerning processing speed in real-world deployments. In response to these challenges, our DRFDCAN has emerged as a promising alternative. It successfully simplifies the network structure, ensuring faster inference while either maintaining or even enhancing image reconstruction quality. The inclusion of a dedicated channel attention block for high-frequency features signifies the model’s adeptness at capturing intricate details. Furthermore, the integration of a more sophisticated residual block underscores our commitment to improving the processing speed without compromising the integrity of the image enhancement. A notable revelation from our research is the pronounced significance of the initial layer features in models that prioritize PSNR. Harnessing this knowledge, we have crafted a distinctive feature extractor, tailored to distill intricate edge and texture details more effectively. The interpretability of deep learning models like DRFDCAN, crucial for clinical trust, remains an area that is ripe for exploration. Adversarial robustness is another concern, as the integrity of medical diagnoses must be safeguarded against potential manipulations. The integration of SR technologies into existing medical workflows, adherence to ethical considerations, and regulatory compliance also pose significant challenges that must be met with deliberate and focused research efforts.

The burgeoning fields of transfer learning and few-shot learning offer promising methods for DRFDCAN to quickly adapt to new tasks, potentially revolutionizing its application to rarer diseases or novel imaging modalities. These approaches could significantly diminish the data requirements for training high-performance SR models, a particularly pertinent advantage in the medical domain, where data availability can be limited due to privacy concerns and regulatory constraints. The proposed DRFDCAN model stands as a significant advancement in medical SR, but fully harnessing its potential will require navigating the complex interplay of technical limitations, ethical considerations, and emerging research directions. Future work should focus on expanding the generalizability and robustness of the model, exploring the ethical landscape of artificial intelligence in healthcare, and ensuring compliance with evolving regulatory standards. The insights from our study thus not only pave the way for the next generation of SR models but also establish a comprehensive roadmap for the responsible and effective integration of these technologies into medical diagnostics.

The evolution of SR, as showcased by our work, signifies a paradigm shift towards models like DRFDCAN. These models prioritize both computational efficiency and unparalleled image reconstruction, which is particularly pivotal in the medical field, where diagnostic accuracy hinges on image clarity. The insights gleaned from our study pave the way for future research, emphasizing the importance of balancing the trifecta of speed, accuracy, and computational demands in the ever-evolving realm of medical super-resolution.

## 6. Discussion

The development and application of the DRFDCAN holds promise for future diagnostic facilities through several significant outcomes. Firstly, the model’s ability to enhance image clarity could revolutionize the precision of diagnostics. This clarity allows for the detection of minute pathological details, which is critical in diagnosing diseases at an early stage when they are most treatable. Moreover, the computational efficiency of DRFDCAN could enable its integration into diagnostic workflows without the need for extensive hardware upgrades. Such efficiency supports the use of SR in a broader range of clinical settings, including those with limited resources. The ability to process images quickly also means that results can be obtained faster, which is crucial in acute medical situations where time is of the essence. If DRFDCAN proves to be adaptable to different imaging modalities, it would be a significant boon, offering a unified SR solution across diverse diagnostic platforms. This flexibility would ensure that high-resolution imaging is not a specialized service but a standard offering, improving diagnostic services across the board. Furthermore, in imaging modalities where radiation is a concern, DRFDCAN could contribute to patient safety. By allowing for lower radiation doses while still maintaining image quality, the model can help in minimizing the risk associated with repeated exposure.

The proposed model’s prowess in image reconstruction is highlighted by its superior PSNR values, which surpass those of established models such as SRCNN [17], RFDN, RDST [32], RFAFN [33], IMDN [34], and LCRCA [35]. For instance, SRCNN [17], an earlier convolutional-neural-network-based model, lags significantly, with a PSNR difference of 1.96 dB compared with the proposed model. This considerable gap underscores the advancements made by the proposed model in producing high-fidelity images. Similar patterns emerge with the SSIM metric, where the proposed model maintains a lead with an impressive 91.56%, indicating its exceptional capability to preserve the structural integrity and texture of the enhanced images. This is particularly noteworthy in comparison to RFAFN [33], which exhibits an SSIM of 83.01%, pointing to potential compromises in structural fidelity. From a computational resource standpoint, the proposed model demonstrates a remarkable efficiency. It consumes only 630 M of memory, which is significantly less than the SRCNN [17] hefty 2387.93 M. This makes the proposed model not only more suitable for environments with limited hardware capabilities but also aligns with the increasing need for energy-efficient AI models, thereby contributing to sustainable computing practices in healthcare. The ability of the DRFDCAN to maintain its performance across various planes and types of images speaks to its robustness and potential for widespread clinical application. As medical imaging technology evolves, such an SR tool could become integral to the development of automated diagnostic systems, enhancing not only image resolution but also facilitating the integration of artificial intelligence in diagnosis, ultimately leading to improved patient outcomes.

## Figures and Tables

**Figure 1 bioengineering-10-01332-f001:**
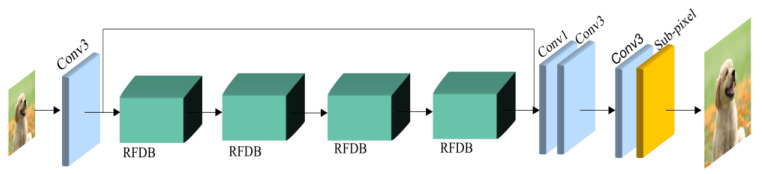
Architecture of baseline model RFDN [13].

**Figure 2 bioengineering-10-01332-f002:**
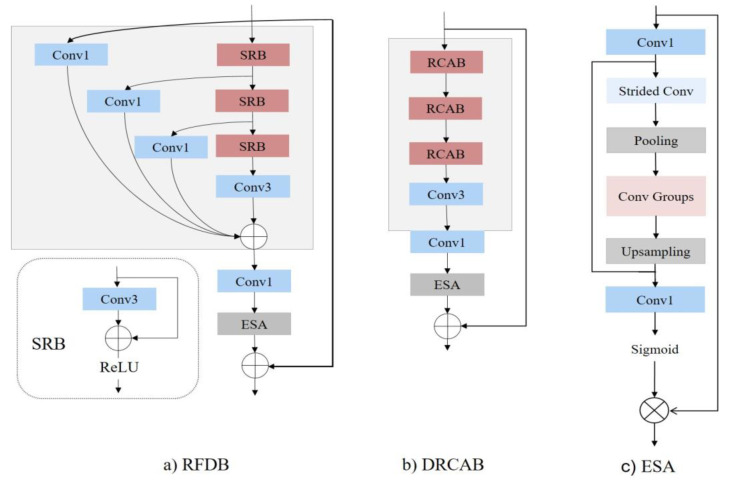
(**a**) RFDB represents the residual feature distillation block. (**b**) DRCAB stands for the residual local feature block. (**c**) ESA is an abbreviation for Enhanced Spatial Attention. In both (**a**,**b**), the number of channels in the output feature maps is indicated adjacent to each layer.

**Figure 3 bioengineering-10-01332-f003:**
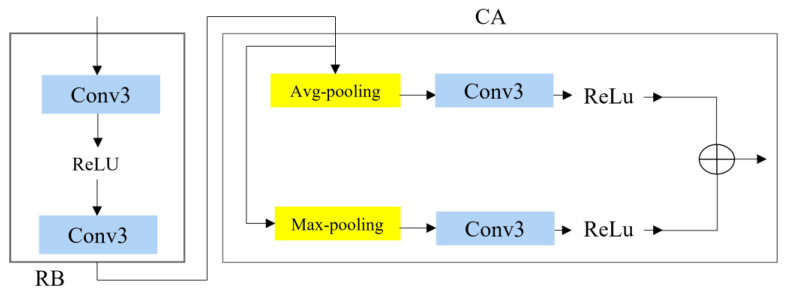
Illustration of RCAB block.

**Figure 4 bioengineering-10-01332-f004:**
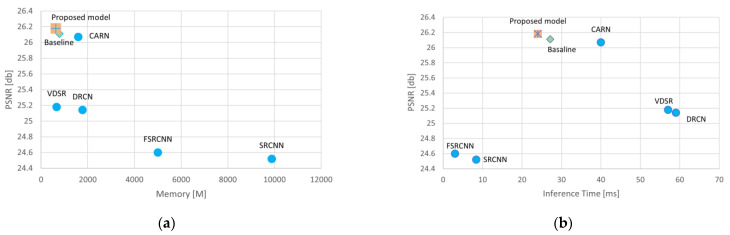
Illustration of PSNR, inference time, and memory consumption of SISR models on the dataset OASIS [37] for scale factor 4×. (**a**) present memory conceptions of baseline model with others. (**b**) present inference time conceptions of baseline model with others.

**Figure 5 bioengineering-10-01332-f005:**
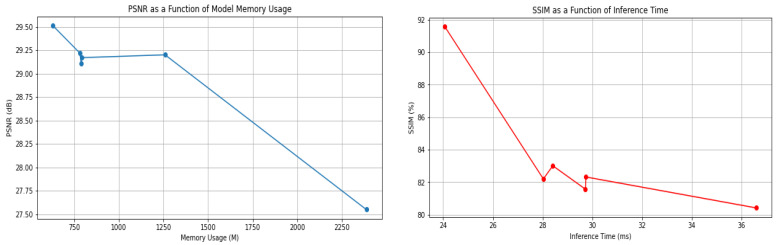
In the provided graphs, we have two plots, each representing the relationship between model performance and computational requirements, one for the PSNR against model memory usage, and the other for SSIM against inference time.

**Figure 6 bioengineering-10-01332-f006:**
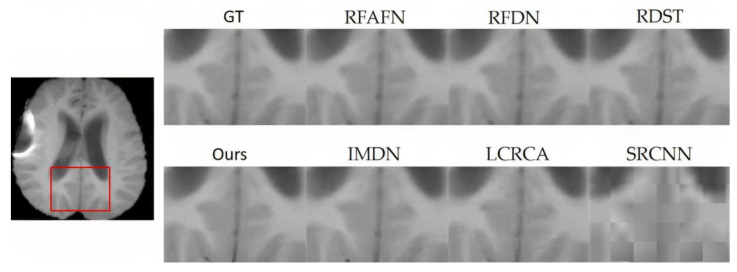
SR outcomes for a randomly selected slice from the BraTS (BraTS: https://www.med.upenn.edu/cbica/brats2020/data.html, accessed on 20 May 2023) testing subset are shown.

**Figure 7 bioengineering-10-01332-f007:**
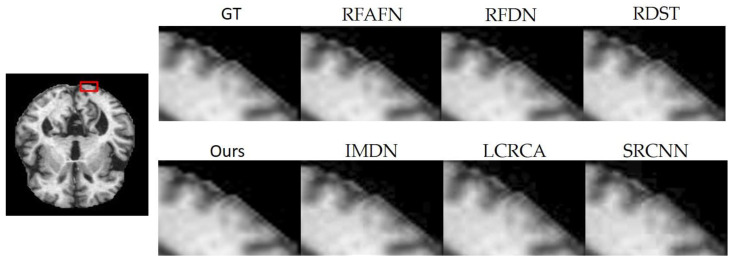
A comparison of the proposed method and state-of-the-art techniques for the ×4 super-resolution task using the OASIS (OASIS: https://www.oasis-brains.org/, accessed on 10 June 2023) dataset is presented.

**Figure 8 bioengineering-10-01332-f008:**
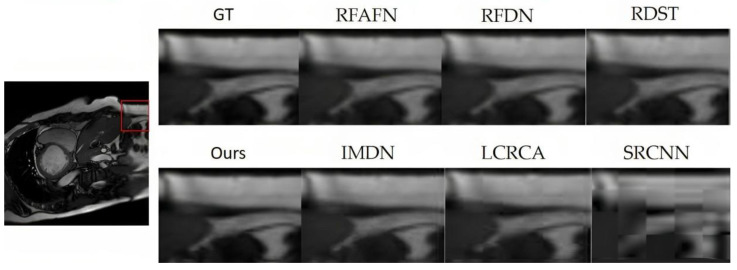
SR outcomes for a randomly chosen slice from the ACDC (ACDC: https://www.creatis.insalyon.fr/Challenge/acdc/databases.html, accessed on 25 June 2023) testing dataset are presented.

**Figure 9 bioengineering-10-01332-f009:**
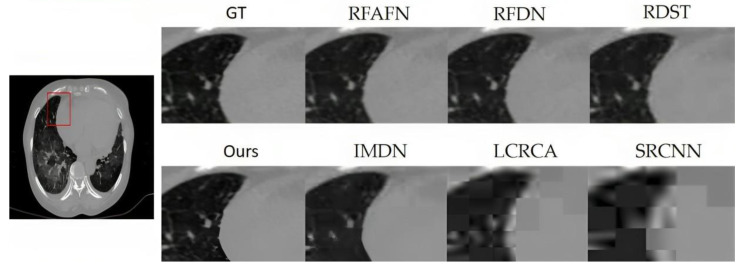
SR findings for a randomly selected slice from the COVID-CT (COVID-19 CT: https://zenodo.org/record/3757476, accessed on 2 June 2023) testing dataset are displayed.

**Table 1 bioengineering-10-01332-t001:** Illustrate comparison of lightweight models based on OASIS dataset.

×4				
Model	PSNR [db]	SSIM	Memory [M]	Inference Time [ms]
**RFDN** [13]	29.11	82.20%	788.19	28.03
**SRCNN** [17]	27.55	80.42%	2387.93	36.58
**RDST** [32]	29.17	82.32%	790.25	29.74
**RFAFN** [33]	29.22	83.01%	780.18	28.41
**IMDN** [34]	29.12	82.23%	856.02	28.07
**LCRCA** [35]	29.16	82.36%	895.36	28.45
**MRI-Net** [34]	29.20	81.57%	1258.81	29.72
** *Proposed model* **	**29.51**	**91.56%**	**630**	**24.07**

**Table 2 bioengineering-10-01332-t002:** The table provides a comprehensive assessment of various super-resolution models evaluated over different datasets and magnifications using the PSNR and SSIM metrics.

×2						
	BraTS		ACDC		COVID-CT	
Mode	PSNR	SSIM	PSNR	SSIM	PSNR	SSIM
**RFDN** [13]	33.65	91.89%	32.84	91.24%	36.87	93.58%
**SRCNN** [17]	30.45	90.20%	32.58	92.15%	35.89	92.65%
**RDST** [32]	33.89	92.27%	33.65	93.41%	36.74	94.87%
**RFAFN** [33]	33.56	92.84%	**35.28**	**94.86%**	37.46	95.45%
**RMISR-BL** [36]	29.46	89.54%	30.08	90.23%	32.89	90.78%
**LCRCA** [35]	32.56	92.11%	31.02	90.84%	35.48	91.99%
**MRI-Net** [34]	31.89	91.57%	31.58	90.99%	32.63	91.14%
** *Proposed model* **	**34.20**	**94.68%**	35.15	94.78%	**37.89**	**96.08%**
**×4**						
	**BraTS**		**ACDC**		**COVID-CT**	
**Model**	**PSNR**	**SSIM**	**PSNR**	**SSIM**	**PSNR**	**SSIM**
**RFDN** [13]	30.11	83.55%	29.21	82.35%	31.87	83.99%
**SRCNN** [17]	28.53	80.89%	27.55	80.24%	28.78	81.24%
**RDST** [32]	30.11	79.58%	27.05	79.23%	33.69	87.99%
**RFAFN** [33]	30.45	81.25%	29.24	80.99%	32.14	89.88%
**RMISR-BL** [36]	30.27	92.68%	27.52	80.41%	31.88	89.12%
**LCRCA** [35]	30.21	81.75%	29.12	82.84%	30.84	82.13%
**MRI-Net** [34]	29.38	81.00%	27.15	79.89%	29.56	82.17%
** *Proposed model* **	**31.55**	**85.12%**	**30.74**	**84.26%**	**34.87**	**89.86%**

## Data Availability

Open Access.

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
