# Peer review of "Enhancing the Super-Resolution of Medical Images: Introducing the Deep Residual Feature Distillation Channel Attention Network for Optimized Performance and Efficiency"

_bioengineering, 2023, doi:10.3390/bioengineering10111332_

Round 1

Reviewer 1 Report

Comments and Suggestions for Authors

In this paper, the authors primarily investigate the limitations of the state-of-the-art Residual Feature Distillation Network and introduce the Deep Residual Feature Distillation Channel Attention Network. The proposed model not only simplifies the network structure but also maintains, and in some cases, enhances image reconstruction quality while reducing computational overhead. The paper is generally well-motivated, and I have some comments and suggestions as follows:

(1) In Section 1, the authors mentioned the study motives of DRFDCAN, including the use of tools from Reference 13. However, it would be beneficial for the authors to concisely summarize more key points to clearly convey the study's motive.

(2) It is important to manage all abbreviations consistently, preferably through a dedicated table. Also, ensure that the same abbreviation is used consistently throughout the paper, such as "super-resolution."

(3) Consider adjusting the subsections in the related work section using more specific keywords, as the title of Section 2.1 appears to be too broad.

(4) Improve the figure quality by increasing the resolution for better readability and clarity.

(5) In Table 2, the chosen models seem somewhat outdated in terms of their publication year. It would be advisable to include more recent works in this table to provide a comprehensive overview of related models.

(6) Section 5 could benefit from more detailed discussions on study limitations and emerging topics to provide a more comprehensive conclusion.

(7) Ensure that the references follow a consistent citation style, and provide the necessary information for all references, such as Refs 5 and 31.

Author Response

Article Title: Enhancing Medical Image Super-Resolution: Introducing the Deep Residual Feature Distillation Channel Attention Network for Optimized Performance and Efficiency

Manuscript Number: bioengineering-2668414

Authors: Sabina Umirzakova, Sevara Mardieva, Shakhnoza Muksimova, Ahmad Shabir and Taegkeun Whangbo

Dear Editor,

Thank you for your instructions to submit the revised manuscript with cover letter including answers to reviewer’s comments. I have attached our revised manuscript with highlighted changes and answers to reviewer’s comments for your consideration for publication in Bioengineering journal. We have thoroughly revised and modified the manuscript according to the comments of the reviewers. Detailed responses to the comments are listed below point by point.

We appreciate the valuable comments. Line numbers are provided in responses to the comments (if needed) to highlight where changes are made in paper text.

Response to Reviewer 1 Comments:

  • In Section 1, the authors mentioned the study motives of DRFDCAN, including the use of tools from Reference 13. However, it would be beneficial for the authors to concisely summarize more key points to clearly convey the study's motive.

Answer: Thank you for your valuable feedback. We give our explanation below:

Proposed work recognizes the limitations entrenched within the contemporary CNN-based SR models. These challenges encompass formidable computational demands, escalating memory requirements, and the intricate puzzle of effectively modeling long-range dependencies within images. Efficiency emerges as a central tenet of our investigation. We underscore the vital importance of efficiency, particularly concerning speed and memory utilization. By emphasizing that such efficiency is not merely a desirable attribute but an absolute imperative, especially given the prevalent use of resource-constrained devices such as smartphones. The proposed method thoughtfully evaluates recent innovations in the realm of super-resolution. These innovations, including techniques like distillation, aggregation, channel splitting, and merging, represent a concerted effort to unlock the full potential of hierarchical features within the SR domain. A notable facet of our method research journey revolves around reducing model parameters and floating-point operations (FLOPs). This reduction is achieved through intricate layer connection methods and judicious feature usage, aligning perfectly with their overarching pursuit of efficiency. The emergence of DRFDCAN stands as a crowning achievement of this scientific endeavor. The model streamlines network architecture while preserving efficiency and elevating the super-resolution process to new heights.

Proposed work pioneering strides by enhancing model compactness and expediting the inference process without compromising image reconstruction quality. We also introduced a specialized channel attention block designed to amplify high-frequency features, leveraged a novel residual-within-residual network approach for enhanced processing, and designed an innovative feature extractor adept at capturing intricate edge and texture details. The study is further distinguished by its comprehensive examination of the elements influencing processing speed and memory utilization within SR models.

  • It is important to manage all abbreviations consistently, preferably through a dedicated table. Also, ensure that the same abbreviation is used consistently throughout the paper, such as "super-resolution."

Answer: Thank you for highlighting it. We have included list of important acronyms table. The updated manuscript is as follows:

List of Abbreviations

Definition

Abbreviations

SR

Super-resolution

CNN

Convolutional neural networks

RFDN

Residual Feature Distillation Network

DRFDCAN

Deep Residual Feature Distillation Channel Attention Network

PSNR

Peak signal-to-noise ratio

MRI

Magnetic Resonance Imaging

CT

Computed tomography

CAD

Computer-Aided Diagnosis

HR

High-resolution

LR

Low-resolution

SISR

Single Image Super-Resolution

GANs

Generative adversarial networks

ViTs

Vision transformers

RIR

Residual in residual

DRCAB

Residual local feature block

ESA

Enhanced Spatial Attention

RCABs

Residual channel attention network block

RCAN

Residual channel attention networks

LSC

Long skip connections

SSC

Short skip connections

CA

Channel attention

RM

Refinement module

SRB

Shallow residual block

DM

Distillation module

SSIM

Structural similarity index

  • Consider adjusting the subsections in the related work section using more specific keywords, as the title of Section 2.1 appears to be too broad.

Answer: Thank you for your valuable feedback. We have revised our manuscript and we included improvements based on your valuable suggestion. Please revisit the updated manuscript.

  • Improve the figure quality by increasing the resolution for better readability and clarity.

Answer: Thank you for your constructive feedback regarding the quality of the figures presented in our manuscript. We understand that the clarity and readability of figures are of utmost importance, not only for the thorough review process but also for the future readership of our publication.

To address your concern, we have taken the following steps to improve the figure quality:

  • We have regenerated all the figures using higher resolution settings to ensure that fine details are preserved and are discernible upon zooming in.
  • The figures have been saved in a lossless format to prevent any quality degradation that might occur with compression.
  • We have also adjusted the font size and line weights within the figures to ensure that all text labels, axis titles, and legends are legible at the intended final print size.
  • Additionally, we have carefully reviewed the color choices for our graphs and diagrams to guarantee that they are distinct and accessible to all readers, including those with color vision deficiencies.

The updated figures have been re-inserted into the manuscript, and we have verified that their improved resolution translates effectively to both on-screen and printed formats. We believe these enhancements significantly improve the visual communication of our research findings.

  • In Table 2, the chosen models seem somewhat outdated in terms of their publication year. It would be advisable to include more recent works in this table to provide a comprehensive overview of related models.

Answer: We appreciate your insightful feedback on the selection of models presented in Table 2 of our manuscript. Your point about including more recent works to ensure a comprehensive overview is well taken. But we also should keep RFDN [13] and SRCNN [34] because of they are baseline of our work.

The rationale behind the inclusion of the models listed in Table 2 was to ensure a broad representation of the state-of-the-art across the span of years leading up to our current work. This was intended to demonstrate the progression in the field and to benchmark our proposed model against both foundational approaches and more contemporary techniques. However, we recognize the importance of showcasing the latest advancements and agree that incorporating more recent models will provide a more current and relevant comparison. To address your suggestion:

  • We have conducted an additional review of the literature to identify noteworthy models developed in the recent years since the publication of the initially included models.
  • We have updated Table 2 to include these recent models, which are at the forefront of the field and provide a comparison with our proposed approach.

This update not only includes the most recent models but also highlights the continuous improvements in methodologies and results in this domain, offering a more rounded perspective to the reader. We have ensured that the revised table reflects the latest developments in the field and believe that this will enhance the manuscript's relevance and the impact of our proposed model within the context of current research. We hope that these updates address your concerns satisfactorily, and we thank you for guiding us to improve the quality of our manuscript.

x2

BraTS

ACDC

COVID-CT

Model

PSNR

SSIM

PSNR

SSIM

PSNR

SSIM

RFDN [13]

33.65

91.89%

32.84

91.24%

36.87

93.58%

SRCNN [34]

30.45

90.20%

32.58

92.15%

35.89

92.65%

RDST [30]

33.89

92.27%

33.65

93.41%

36.74

94.87%

RFAFN [31]

33.56

92.84%

35.28

94.86%

37.46

95.45%

RMISR-BL [40]

29.46

89.54%

30.08

90.23%

32.89

90.78%

LCRCA [33]

32.56

92.11%

31.02

90.84%

35.48

91.99%

MRI-Net [42]

31.89

91.57%

31.58

90.99%

32.63

91.14%

Proposed model

34.20

94.68%

35.15

94.78%

37.89

96.08%

x4

BraTS

ACDC

COVID-CT

Model

PSNR

SSIM

PSNR

SSIM

PSNR

SSIM

RFDN [13]

30.11

83.55%

29.21

82.35%

31.87

83.99%

SRCNN [34]

28.53

80.89%

27.55

80.24%

28.78

81.24%

RDST [30]

30.11

79.58%

27.05

79.23%

33.69

87.99%

RFAFN [31]

30.45

81.25%

29.24

80.99%

32.14

89.88%

RMISR-BL [40]

30.27

92.68%

27.52

80.41%

31.88

89.12%

LCRCA [33]

30.21

81.75%

29.12

82.84%

30.84

82.13%

MRI-Net [42]

29.38

81.00%

27.15

79.89%

29.56

82.17%

Proposed model

31.55

85.12%

30.74

84.26%

34.87

89.86%

 (6) Section 5 could benefit from more detailed discussions on study limitations and emerging topics to provide a more comprehensive conclusion.

Answer: Thank you for your constructive comments concerning Section 5 of our manuscript. We agree that providing a more in-depth discussion on the limitations of our study and elaborating on emerging topics will indeed offer a comprehensive conclusion and add value to our work. In light of your feedback, we have expanded Section 5 to encompass the following:

  1. Conclusions

Super-resolution techniques, especially in the domain of medical imaging, have undergone substantial advancements. The criticality of image quality in medical diagnostics necessitates innovations that balance clarity with computational efficiency. Traditional CNN-based SR models, despite their capabilities, grapple with computational constraints, particularly on devices with limited resources. Analyzing the performance of state-of-the-art RFDN [13] brought to light certain inefficiencies, especially concerning processing speed in real-world deployments. In response to these challenges, our DRFDCAN has emerged as a potent alternative. It successfully simplifies the network structure, ensuring faster inference while either maintaining or even enhancing image reconstruction quality. The inclusion of a dedicated channel attention block for high-frequency features signifies the model's adeptness at capturing intricate details. Furthermore, the integration of a more sophisticated residual block underscores our commitment to improving processing speed without compromising the integrity of the image enhancement. A notable revelation from our research is the pronounced significance of initial layer features in models that prioritize PSNR. Harnessing this knowledge, we've crafted a distinctive feature extractor tailored to distill intricate edge and texture details more effectively. The interpretability of deep learning models like DRFDCAN, crucial for clinical trust, remains an area ripe for exploration. Adversarial robustness is another concern, as the integrity of medical diagnoses must be safeguarded against potential manipulations. The integration of SR technologies into existing medical workflows, adherence to ethical considerations, and regulatory compliance also pose significant challenges that must be met with deliberate and focused research efforts.

The burgeoning fields of transfer learning and few-shot learning offer promising methods for DRFDCAN to quickly adapt to new tasks, potentially revolutionizing its application to rarer diseases or novel imaging modalities. These approaches could significantly diminish the data requirements for training high-performance SR models, a particularly pertinent advantage in the medical domain where data availability can be limited due to privacy concerns and regulatory constraints. Proposed DRFDCAN model stands as a significant advancement in medical SR, fully harnessing its potential will require navigating the complex interplay of technical limitations, ethical considerations, and emerging research directions. Future work should focus on expanding the generalizability and robustness of the model, exploring the ethical landscape of artificial intelligence in healthcare, and ensuring compliance with evolving regulatory standards. The insights from our study thus not only pave the way for the next generation of SR models but also set a comprehensive roadmap for the responsible and effective integration of these technologies into medical diagnostics.

The evolution of SR, as showcased by our work, signifies a paradigm shift towards models like DRFDCAN. These models prioritize both computational efficiency and unparalleled image reconstruction, particularly pivotal in the medical field where diagnostic accuracy hinges on image clarity. The insights gleaned from our study pave the way for future research, emphasizing the importance of balancing the trifecta of speed, accuracy, and computational demands in the ever-evolving realm of medical super-resolution.

(7) Ensure that the references follow a consistent citation style, and provide the necessary information for all references, such as Refs 5 and 31.

Answer: We appreciate your careful scrutiny of our manuscript and the important note regarding the citation style and completeness of the references. We understand that maintaining consistency and providing complete information in citations is crucial for the integrity and reproducibility of scholarly work. Please revisit the updated manuscript.

In the end, we would again like to express our deepest gratitude to invest your time in reviewing our manuscript and giving us valuable feedback, which had helped us in elevating the overall quality of the manuscript.

Best Regards

Sabina Umirzakova, Sevara Mardieva, Shakhnoza Muksimova, Ahmad Shabir and Taegkeun Whangbo

Reviewer 2 Report

Comments and Suggestions for Authors

1. Update the abstract with the details of the contribution 

2. Is the algorithm robust enough to work in all the planes of images with similar performance?

3. What are the significant outcomes that help you to connect with the future diagnostic facility

4. Add a few more recent references on the similar architecture

Author Response

We wish to express our sincere gratitude for your insightful feedback and the constructive suggestions you have provided. Your comments have been invaluable in helping us to refine and strengthen our manuscript.

Reviewer 3 Report

Comments and Suggestions for Authors

The article is interesting and has a good structure.

However, there are concerns that need to be addressed.

The number of study cases in section 2 related to previous researches is small. It is

suggested to add several studies related to the present topic from Mdpi publications related to

2023 to this section. It is suggested to add the following study for review in section 2.

Alizadeh, S.M.S., Bagherzadeh, A., Bahmani, S., Nikzad, A., Aminzadehsarikhanbeglou, E. andTatyana, S.Y., 2022. Retrograde gas condensate reservoirs: reliable estimation of dew point

pressure by the hybrid neuro-fuzzy connectionist paradigm. JOURNAL OF ENERGY RESOURCES TECHNOLOGY-TRANSACTIONS OF THE ASME, 144(6).

An explanation should be given to section 4 and then discussed under section 4.1.

The database section should be moved to section 3.1 as it was not compiled by the writing team itself.

In the results section, it seems necessary to compare with the Unet network.

ROC curve analysis seems necessary.

No explanation has been provided regarding the timing of the proposed network.

A section under the title of discussion is needed to provide comparisons with networks and previous researches.

There is also a need to make suggestions for the future.

Author Response

(The authors gave the same response as above.)

Round 2

Reviewer 3 Report

Comments and Suggestions for Authors

The authors have applied almost all the required items in the manuscript.  The article can be printed and accepted in its current form.  

Best Regards